# Overexpression of the *QKI* Gene Promotes Differentiation of Goat Myoblasts into Myotubes

**DOI:** 10.3390/ani13040725

**Published:** 2023-02-17

**Authors:** Sijia Chen, Shu Niu, Wannian Wang, Xiang Zhao, Yangyang Pan, Liying Qiao, Kaijie Yang, Jianhua Liu, Wenzhong Liu

**Affiliations:** Department of Animal Genetics, Breeding and Reproduction, College of Animal Science, Shanxi Agricultural University, Jinzhong 030801, China

**Keywords:** gene, goat, isoform, myoblast, *QKI*, RNA-Seq, skeletal muscle

## Abstract

**Simple Summary:**

Meat goats are highly valued as excellent sources of lean dietary muscle-protein. However, the molecular mechanisms by which muscle cells and tissues are generated and the genes and signaling pathways regulating this process in meat goats are poorly understood. The present study applied currently conventional molecular-genetics methods and determined that the major isoform of the *QKI* gene drive the conversion of embryonic myoblast cells into mature muscle fiber and tissue. These genes activate other genetic factors and signaling pathways that are all implicated in the differentiation of muscle progenitor cells and the formation of mature, differentiated muscle tissue. Although future investigations will be necessary to elucidate the entire molecular mechanism involved, we believe that the discoveries reported herein will facilitate the genetic engineering of novel goat lines with enhanced musculature and, by extension, superior meat yield and quality, breeder profitability, and consumer satisfaction.

**Abstract:**

The *QKI* genes encode RNA-binding proteins regulating cell proliferation, differentiation, and apoptosis. The Goat *QKI* has six isoforms, but their roles in myogenesis are unclear. In this study, the six isoforms of the *QKI* gene were overexpressed in goat myoblast. Immunofluorescence, qPCR and Western blot were used to evaluate the effect of *QKI* on the differentiation of goat myoblast. An RNA-Seq was performed on the cells with the gain of the function from the major isoforms to screen differentially expressed genes (DEGs). The results show that six isoforms had different degrees of deletion in exons 6 and 7, and caused the appearance of different types of encoded amino acids. The expression levels of the *QKI-1* and *QKI-5* groups were upregulated in the biceps femoris and latissimus dorsi muscle tissues compared with those of the *QKI-4*, *QKI-7*, *QKI-3* and *QKI-6* groups. After 6 d of myoblast differentiation, *QKI-5* and the myogenic differentiators *MyoG*, *MyoD*, and *MyHC* were upregulated. Compared to the negative control group, *QKI* promoted myotube differentiation and the myoblasts overexpressing *QKI-5* formed large, abundant myotubes. In summary, we identified that the overexpression of the *QKI* gene promotes goat-myoblast differentiation and that *QKI-5* is the major isoform, with a key role. The RNA-Seq screened 76 upregulated and 123 downregulated DEGs between the negative control and the *QKI-5*-overexpressing goat myoblasts after d 6 of differentiation. The GO and KEGG analyses associated the downregulated DEGs with muscle-related biological functions. Only the pathways related to muscle growth and development were enriched. This study provides a theoretical basis for further exploring the regulatory mechanism of *QKI* in skeletal-muscle development in goats.

## 1. Introduction

Meat-production performance is an economically important trait in meat goats (*Capra hircus*) and it is a research focus in genetic breeding. Lingqiu gray-backed goats are a typical local breed in Shanxi Province, China. They have several excellent production traits, such as strong adaptability and good meat quality. Meat yield and quality are closely related to skeletal-muscle growth and development. Hence, muscle growth and development are regarded as the bases of meat-production performance in livestock and poultry. Elucidating the mechanisms regulating goat-muscle-tissue development will allow the identification of their molecular-level effects on meat quality and production performance and lay the foundation for future genetic meat-goat breeding. Skeletal-muscle growth and development are highly coordinated and includes myoblast proliferation, differentiation, migration, and fusion [1]. These processes entail a complex regulatory-gene-expression network and function mainly through the precise regulation of intercellular signaling and specific gene expression [2]. They are regulated by myogenic regulatory factors (MRFs) and certain transcription factors (TFs) [3] and include protein synthesis and degradation pathways, as well as factors implicated in myogenic cell fusion [1,4]. The RNA-binding proteins (RBPs) regulate muscle growth and development by interacting with coding and non-coding RNAs [5].

The RBPs affect RNA structure and interactions by binding RNA. They participate in RNA transcription, processing, degradation and nuclear export. Thus, they regulate biological RNA function [6,7,8]. These RBPs are key RNA-processing regulators in skeletal muscles. The misregulation of RBP expression, localization, or function causes defects in mRNA stabilization, polyadenylation, and alternative splicing (AS) [9]. In AS, pre-mRNA is spliced in various ways to form mRNA-splicing isoforms with different sequences. In this manner, a single gene can generate multiple mRNA and protein isoforms, thereby increasing protein-expression diversity [10]. The RBPs regulate changes in AS by altering their expression levels during development. Most of the splicing regulators identified to date are RBPs [11]. The RBP known as quaking homolog, KH domain RNA binding (*QKI*), is a member of the STAR (signal transduction and activation of RNA) family. The deletion of the upstream regulatory region of *QKI* results in tissue-specific *QKI* downregulation and leads to neuromyelin-development disorders. Mice with this genetic defect present with a trembling phenotype [12]. The expression level of *QKI* gene increases during myogenesis and regulate alternative splicing by binding the ACUAA motifs that are enriched in regions downstream of muscle-specific exons. When *QKI* binds upstream of the alternative splicing region, it promotes inclusion [13]. When it binds downstream of the alternative splicing region, however, it promotes skipping [14]. The *QKI* proteins are localized on mouse and human chromosomes 17 and 6, respectively; they encode the isoforms *QKI-5*, *QKI-6* and *QKI-7*, which share the same K homology (KH) structural domain [15,16]. In mouse myoblasts, different variants display a network of autoregulating and cross-regulating *QKI* protein isoforms [17]. The *QKI* is a global splicing regulator during C2C12 muscle-cell differentiation [14]. It is expressed in differentiating slow muscle cells [18]. It also participates in myogenesis by regulating alternative splicing [14], early myofibrillar formation [19] and circular RNA formation [20]. Hence, *QKI* is directly or indirectly involved in skeletal-muscle growth and development.

To the best of our knowledge, the mechanism by which the RBP *QKI* regulates goat skeletal-muscle-myoblast differentiation has never been reported. Although it is known that goat *QKI* has six isoforms, it is unclear which of these is upregulated in goats muscle. The aim of the present study, then, was to identify the major *QKI* isoforms by investigating their expression trends during goats-myoblast differentiation and to determine their regulatory effects on myogenesis. In addition, in order to further explore the molecular mechanism of *QKI* in goat-myoblast differentiation, RNA-seq was performed to screen for differentially expressed genes (DEGs). The discoveries reported herein could provide theoretical and practical bases for the regulation of skeletal-muscle growth and development in meat goats.

## 2. Materials and Methods

### 2.1. Goat-Myoblast Isolation and Culture

All animal experiments were performed according to the animal procedures established by the Ministry of Agriculture of China and were approved by the Institutional Animal Care and Use Committee of Shanxi Agricultural University. Biceps femoris (BF) and longissimus dorsi (LD) muscle tissue was collected from Lingqiu gray-backed goats ≤ 1 mo old. One part of each tissue was isolated and cultured while the other was used for total RNA extraction. Myoblasts were isolated and cultured in Dulbecco’s modified Eagle’s medium (DMEM) supplemented with 10% (*v*/*v*) fetal bovine serum (FBS, Gibco, Waltham, MA, USA) and 1% (*w*/*v*) penicillin/streptomycin (Gibco, Waltham, MA, USA). When the myoblasts reached 90% confluence, the growth medium was replaced with differentiation medium consisting of DMEM, 2% (*v*/*v*) horse serum (Invitrogen, Carlsbad, CA, USA), and 1% (*w*/*v*) penicillin/streptomycin. The medium was changed every 2 d.

### 2.2. Total-RNA Extraction and cDNA Synthesis

Total RNA was extracted from the goat BF tissues with TRIzol reagent. The RNA concentration was measured by nucleic-acid-protein assay (Bio-Rad Laboratories, Hercules, CA, USA). The extracted RNA was reverse-transcribed to cDNA with a PrimeScript^®^ RT reverse transcription kit (Thermo Fisher Scientific, Waltham, MA, USA).

### 2.3. Primer Design and Overexpression Vector Construction

A search of the NCBI database (https://www.ncbi.nlm.nih.gov, accessed on 1 April 2022) revealed six predicted goat *QKI* isoforms, namely, *QKI-1*, *QKI-3*, *QKI-4*, *QKI-5*, *QKI-6* and *QKI-7*. A sequence alignment showed that the sequences of *QKI-1* and *QKI-5*, of *QKI-3* and *QKI-6* and of *QKI-4* and *QKI-7* differed by 24 bases in the center of the coding region. All other sequences were identical. The qPCR primer pairs *QKI-15*, *QKI-36* and *QKI-47* were designed with Primer Premier v. 6.0 (Premier Biosoft, San Francisco, CA, USA) (Table 1). The sequences of the CDS regions of the six *QKI* isoforms were sent to Sangon Biotech Co., Ltd. (Shanghai, China) for full gene synthesis to obtain overexpression vectors for the six *QKI* isoforms. The vector was pHBLV-CMVIE-ZsGreen-T2A-puro and the 5′ and 3′ digestion sites were *Eco*RI and *Xba*I, respectively.

### 2.4. Cell Transfection and Induced Differentiation

The myoblasts were cultured in six-well plates (Corning, New York, NY, USA) before transfection. The goat myoblasts were used for cell transfection when they reached ~60% confluence. Before 24 h of transfection, 293T cells were inoculated into 10 cm cell-culture dishes (Corning, USA) and co-transfected with recombinant plasmids and the packaging plasmids, pMD2G and psPAX2, via Lipofectamine^TM^ 3000 transfection reagent (Thermo Fisher Scientific). The existing medium was replaced with fresh complete medium after 6 h of transfection. Cell cultures were collected after 48 h and 72 h transfection, passed through a 0.22-micrometer filter and centrifuged at 72,000× *g* and 4 °C for 2 h. The supernatant was discarded and the pellet was resuspended in 500 μL fresh culture medium to obtain the viral fluid. Successful infection of the goat myoblast by the viral fluid was observed under a fluorescent microscope (Nikon, Tokyo, Japan) and appeared as green fluorescence. When the cells reached 90% confluence, they were subjected to cell-differentiation medium containing 2% (*v*/*v*) horse serum. That time point was recorded as day 0 of differentiation. The cell-differentiation medium was changed every 2 d. Total RNA and protein were collected from the cells at days 0 and 6 of differentiation. Cell morphology and myotube appearance were observed under a microscope.

### 2.5. Immunofluorescence

After 6 d of differentiation, the cells were washed twice with phosphate-buffered saline (PBS), fixed in cold methanol for 10 min, permeabilized with 0.1% (*v*/*v*) Triton X-100 for 10 min, blocked with 5% (*v*/*v*) bovine serum albumin (BSA) and incubated with primary antibodies (anti-desmin polyclonal antibody) at 4 °C overnight. The cells were then washed thrice with PBS, stained with fluorophore-conjugated secondary antibodies (CoraLite594-conjugated goat anti-rabbit immunoglobulin G (IgG) (H + L) 1:400) at room temperature for 1 h and incubated with 4′,6-diamidino-2-phenylindole (DAPI) for 10 min. Images of the cells were captured with a DMi8 microscope (Leica Microsystems GmbH, Wetzlar, Germany). The number of fusion cells per unit area of differentiated myotubes was analyzed. The fusion indices (% of nuclei in the myotubes) were determined with ImageJ 2.3.0 (National Institutes of Health, Bethesda, MD, USA).

### 2.6. Western Blotting

Protein was extracted from the myoblast with radioimmunoprecipitation assay (RIPA) lysis buffer on ice for 30 min. The protein fractions were collected by centrifugation at 10,000× *g* and 4 °C for 10 min, separated by sodium dodecyl sulfate-polyacrylamide gel electrophoresis (SDS-PAGE) and transferred to nitrocellulose membranes with a Trans-Blot turbo transfer system (Bio-Rad Laboratories). The membranes were blocked with 5% (*w*/*v*) nonfat dry milk in PBS and immunoblotted with the primary antibodies anti-*QKI-5* (1:1500; Proteintech, Rosement, IL, USA) and anti-glyceraldehyde-3-phosphate dehydrogenase (GAPDH) (1:6000; Bioss, Beijing, China) at 4 °C overnight. The membranes were then washed thrice in Tris-buffered saline plus Tween 20 (TBST) and incubated with secondary anti-IgG antibody (1:1000; LI-COR Biosciences, Lincoln, NE, USA) at room temperature for 1 h. Staining was performed in an Odyssey infrared imaging system (LI-COR Biosciences, Lincoln, NE, USA).

### 2.7. RNA Extraction, Library Construction, and Transcriptome Sequencing

Goat myoblasts from the *QKI-5*-overexpressing and negative control groups were collected at 6 d of differentiation. Total RNA was extracted with TRIzol reagent (Takara, Dalian, China) following the manufacturer’s protocol. The RNA integrity was evaluated with the RNA Nano 6000 assay kit and a Bioanalyzer 2100 system (Agilent Technologies, Santa Clara, CA, USA). Total RNA from each sample was used in Illumina sequencing at Novogene Bioinformatics Technology Co., Ltd. (Beijing, China). The mRNA was purified from the total RNA using poly-T oligo-attached magnetic beads. To select cDNA fragments 370–420 bp in length, the library fragments were purified in an AMPure XP system (Beckman Coulter, Beverly, MA, USA). A PCR was then performed using Phusion high-fidelity DNA polymerase, universal PCR primers and index (X) primer. The PCR products were purified in the AMPure XP system and library quality was assessed in the Agilent Bioanalyzer 2100 system.

The index-coded samples were clustered in a cBot cluster-generation system with a TruSeq PE cluster kit v3-cBot-HS (Illumina, San Diego, CA, USA) according to the manufacturer’s instructions. The library preparations were then sequenced on an Illumina NovaSeq platform (Illumina) and 150-base-pair paired-end reads were generated.

### 2.8. Quality Control and Sequence Alignment

Raw reads in fastq format were processed in fastp v. 0.19.7 (https://github.com/mskcc-cwl/fastp_0.19.7, accessed on 21 October 2022). Clean reads were obtained by removing reads containing adapter or poly-N and low-quality reads. The Q20, the Q30 and the GC content of the clean data were calculated. The reference genome index was built with HiSat2 v. 2.0.5 (http://daehwankimlab.github.io/hisat2/, accessed on 22 October 2022). Paired-end clean reads were aligned to the goat reference genome. The reference genome and the gene-model-annotation files were directly downloaded from the genome website (http://ftp.ensembl.org/pub/release-99/fasta/capra_hircus/, accessed on 22 October 2022).

### 2.9. Gene-Expression Quantification and Analysis of Differentially Expressed Genes (DEGs)

FeatureCounts v. 1.5.0-p3 (https://github.com/byee4/featureCounts, accessed on 24 October 2022) was used to count the reads mapped to each gene. The fragments per kilobase of transcript per million mapped reads (FPKM) was calculated for each gene based on the length of the gene and the read counts mapped to it. Analysis of differentially expressed genes was performed using the DESeq2 v. 1.20.0 package in R (R Core Team, Vienna, Austria). The *p*-values were subjected to the Benjamini and Hochberg correction to control for the false-discovery rate (FDR). Genes with adjusted *p*-value ≤ 0.05 were deemed differentially expressed.

### 2.10. GO and KEGG Enrichment Analyses of DEGs

Gene ontology (GO) enrichment analysis of the DEGs was performed using the clusterProfiler package in R and the gene-length bias was corrected. The GO terms with corrected *p* < 0.05 were deemed significantly enriched by the DEGs. The clusterProfiler package in R was also used to test for statistical enrichment of the DEGs in the KEGG pathways.

## 3. Results

### 3.1. Sequencing Analyses of Various Goat QKI Isoforms

We compared the coding sequence (CDS) nucleotides of the goat *QKI* isoforms obtained from NCBI (Figure 1A). We found that the sequences for *QKI-1* and *QKI-5*, *QKI-3* and *QKI-6* and *QKI-4* and *QKI-7* differed by 24 bases in the center of the coding region, whereas the rest of the sequences were identical. We submitted the nucleotide sequences obtained from NCBI to a goat-wide genome database. The NCBI open reading frames (ORFs) were used to translate the base sequence in the coding region of each isoform into an amino-acid sequence. We then compared the structures of the different splice variants, as shown in Figure 1B. The alignment results for the coding region showed that the first five exons were identical for all six *QKI* isoforms. At the exon-6-translation starting position, *QKI-1*, *QKI-3*, and *QKI-4* lost eight amino acids, but their amino-acid domains were unchanged. Starting from the exon-7-amino-acid sequence, the amino-acid structure of each isoform underwent certain changes relative to *QKI-1* and *QKI-5*. The *QKI-3* and *QKI-6* were missing 16 amino acids, while the *QKI-4* and *QKI-7* were missing 22 amino acids. These changes caused different types of amino acid to be encoded.

### 3.2. Expression Analysis of Different QKI Isoforms in Goats-Muscle Cells and Tissues

The qPCR showed that the *QKI* was upregulated in the BF muscle compared with the LD muscle. The expression was significantly increased in the *QKI-1*, *QKI-5* group relative to the *QKI-4, QKI-7* and *QKI-3, QKI-6* groups (Figure 2A). The transcript-expression trends during the myoblast differentiation were determined by the overexpression of all six *QKI* isoforms in the BF myoblasts. The expression level of the *QKI-15* group was significantly higher than those of the other isoforms during the myoblast differentiation. The highest expression level was determined for d 6 of the differentiation (Figure 2B).

### 3.3. QKI Regulation of Myoblast Myogenesis and Identification of the Major QKI Isoform

The various *QKI* isoforms were overexpressed during the myoblast differentiation for 6 days. The myotubes were subjected to immunofluorescence to compare the relative myoblast differentiation. The myoblasts transfected with *QKI-5* overexpression formed larger and more numerous myotubes than those transfected with the negative control (NC) (Figure 2C). The fusion indices of the myotubes differentiated by *QKI-5* was significantly higher than those of the myotubes differentiated by the other *QKI* isoforms (Figure 2D). Hence, *QKI-5* was the major isoform implicated in goat-myoblast differentiation. Furthermore, the qPCR showed that the expression levels of the key myogenic genes, *MyoG*, *MyoD*, and *MyHC*, significantly increased in response to the *QKI-5* overexpression (Figure 2E). The *QKI-5* overexpression also increased the *QKI-5* protein abundance relative to the NC (Figure 2F).

### 3.4. Summary of RNA-Seq Data

The transcriptome sequencing of all the samples showed that there were ≥5.82 G of clean reads after the original data filtering, sequencing-error-rate examination, and GC content distribution. The Q30 scores for the clean bases were all >92.29% in all six samples (Appendix A). The clean reads of each sample were compared against the goat reference genome. The total number of fragments after sequencing-data-quality control (QC) was in the range of 38,779,502–46,014,404 and ~96.19–96.6% of the reads were mapped. About 92.09–92.79% of the reads were uniquely mapped to the goat reference genome. The mapped reads were evenly distributed between the positive and negative genome strands (Appendix A).

### 3.5. Differentially Expressed Gene (DEG) Analysis

The EBSeq algorithm (|log2(FoldChange)| > 1 and padj < 0.05) disclosed 199 DEGs between the OE and NC groups. Of these, 76 were upregulated and 123 were downregulated. *ACTA1*, *CAV3*, *TNNC1*, *TNNC2*, *TNNI1*, *TNNT1* and *TNNT3* were highly significantly differentially expressed in relation to muscle growth and development. The distribution of the DEGS in the two treatment groups is displayed in a volcano plot (Figure 3A). The DEGs were selected and pooled as gene sets for cluster analysis. The genes in the heatmap with similar expression patterns were clustered together. Figure 3B shows that three repeats had highly similar gene-expression levels and patterns between groups, whereas different gene groups had distinct patterns. Thus, both sample groups had good repeatability and clustering effects.

### 3.6. Gene Ontology (GO) and Kyoto Encyclopedia of Genes and Genomes (KEGG) Pathway Analyses of DEGs

The DEGs were annotated according to the molecular-function, cellular-component, and biological-process GO categories. The 10 most significantly enriched pathways in each function are shown in Figure 4A (Appendix A). For the molecular-function terms, the DEGs were enriched mainly in binding functions, such as cytoskeletal protein binding, actin binding, and structural constituents. For the cellular-component terms, the DEGs were enriched mainly in myofibril. For the biological-process terms, the DEGs were enriched mainly in muscle-structure development, muscle-tissue and organ development, and muscle-cell development. We also selected 10 GO terms directly involved in muscle growth and development, and obtained 20 DEGs associated with them (Table 2).

We used the KEGG database for DEG pathway annotation. Twenty-five pathways were significantly (*p* < 0.05) enriched among all the DEGs, including hypertrophic cardiomyopathy (HCM), dilated cardiomyopathy (DCM), cardiac-muscle contraction, focal adhesion, and MAPK signaling (Figure 4B).

## 4. Discussion

Muscle growth and development are important indicators of goat performance and meat quality. Understanding muscle development is crucial in agricultural-animal breeding, livestock and poultry genetic-resource conservation, and the diagnosis and treatment of different muscle-related diseases. Physiological changes in skeletal-muscle development are accompanied by numerous transcriptional and posttranscriptional changes, including those controlled by alternative splicing [21]. The latter process is widespread among eukaryotes, regulates gene expression and creates proteome diversity. Alternative splicing events are particularly complex in higher eukaryotes [22], as more advanced organisms tend to have relatively abundant genetic isoforms [23]. The latter are mainly produced when genes and pre-mRNA are transcribed. Exons generate different gene sequences by cutting, arranging, combining and relinking. These transformations are translated into various protein constructs. Alternative splicing is controlled mainly by RNA-binding proteins (RBPs) that change their expression levels during development. The RBP *QKI* directly acts on pre-mRNA and regulates mRNA and protein synthesis through different splicing forms [24]. It also affects mRNA and protein expression during posttranscriptional transport, localization, and stabilization [25].

Here, we identified the mRNA-level tissue expression and localization of six *QKI* isoforms. We investigated the expression trend in *QKI* isoforms during myoblast differentiation. This research provides a theoretical basis for elucidating the mechanisms by which goat *QKI* regulates myoblast formation. We found that the myoblasts gradually differentiated into myotubes with a prolongation of the differentiation time up until 6 days. After this time, no further increase in differentiation was detected. The qPCR showed that *QKI* was significantly upregulated in the *QKI-1*, *QKI-5* group compared with the *QKI-4*, *QKI-7* and *QKI-3*, *QKI-6* groups. This finding was consistent with the trend observed in the myoblast differentiation. Therefore, *QKI-1* and *QKI-5* may be the most highly expressed isoforms during goat-myoblast differentiation and could facilitate subsequent determinations of major isoforms. We then overexpressed the six *QKI* isoforms in the myoblasts, induced myogenic differentiation, stained the myotubes, and microscopically examined the relative differences in myoblast differentiation. The fusion index of the myotubes differentiated by *QKI-5* was significantly higher than those of the myotubes differentiated by the other isoforms. Hence, *QKI-5* plays a key role in myoblast differentiation. These results indicated that *QKI-5* is a major isoform implicated in goat-myoblast differentiation. This discovery is consistent with the results of some previous studies. The *QKI-5* automatically regulates its own alternative splicing through a self-splicing mechanism and was described as a major regulator of RNA metabolism in oligodendrocytes [16]. In lung cells, *QKI-5* is the dominant isoform at the mRNA and protein levels. It suppresses tumorigenesis by inhibiting *ADD3*-exon-14 splicing [26]. Furthermore, it acts as a nuclear localization signal and regulates pre-mRNA splicing [27,28]. In addition, it is the main isoform in human embryonic stem cells (hESCs) and cardiogenic progenitor cells during heart development [29]. Myoblasts transfected with *QKI* overexpression formed relatively large and abundant myotubes compared to cells transfected with NC. Thus, *QKI* may enable myoblasts to differentiate into myotubes. Our Western blotting showed that the protein abundance of the overexpressed *QKI-5* was significantly higher than that of the control. Our qPCR demonstrated that the myogenic differentiation markers *MyoG*, *MyoD* and *MyHC* were significantly upregulated. Therefore, *QKI* promotes goat-myoblast differentiation.

To clarify the molecular mechanism by which *QKI* promotes goat-myoblast differentiation, we screened significantly differentially expressed genes (DEGs) between the experimental and control groups. The use of RNA high-throughput sequencing (RNA-Seq) is indispensable in the analysis of differential gene expression and alternative mRNA splicing at the transcriptome level. Here, we applied second-generation high-throughput sequencing on the NC goat myoblasts and those overexpressing *QKI-5*. We found that the Q30 was >92% and the GC base content was in the range of 51.32–52.33%. The data quality is considered satisfactory when Q30 > 80% and the GC base content is in the range of 50–60%. To determine which genes and sequenced fragments are transcribed, quality-controlled clean reads must be compared against a reference genome. Here, >92% of the clean reads were successfully mapped to the goat genome. Hence, the data were adequate for the subsequent analysis.

In the present study, we attempted to identify the key genes and signaling pathways related to muscle growth and development. A gene ontology (GO) analysis showed that most of the DEGs regulated muscle function in general and muscle contraction in particular. Thus, *QKI* may be related to muscle-tissue development. There were far more downregulated than upregulated DEGs enriched in biological function, including muscle contraction and skeletal-muscle development. Hence, *QKI* may participate upstream of the muscle-development signaling pathway. The genes downstream of *QKI* might be critical for muscle growth and development. The proteins *TNNC1*, *TNNC2*, *TNNI1* and *TNNT3* are all troponins (Tn). Furthermore, TnC, TnI and TnT comprise Tn, which collaborates with tropomyosin to regulate muscle contraction. The Tn complexes to form a calcium-sensitive molecular switch on striated muscle filaments. It regulates striated muscle contraction in response to changes in intracellular calcium concentration [30] and affects the meat quality of livestock [31,32]. In mice, *TNNT3* is vital for growing muscle and C2C12 expression and the alternative spliceosomes that gradually increase with myotube differentiation. Therefore, *TNNT3* performs important biological functions in muscle growth and development [33]; *TNNC2* plays a critical role in skeletal muscle contraction. The *TNNC2* gene is expressed during the myoblast differentiation and skeletal muscle development [34]; *TNNI1* genes are switched on during skeletal-muscle myogenesis and are expressed during the early periods of development [35]. These results suggest that the expression of genes related to muscle-tissue development are downregulated after overexpressing the *QKI* gene in myoblasts, which suggests that the *QKI* gene can inhibit the expression of genes related to muscle development. Future research should elucidate the mechanism of the DEGs associated with muscle growth and development.

The KEGG analysis helped to identify the 20 most significantly enriched pathways related to the DEGs, including hypertrophic cardiomyopathy, dilated cardiomyopathy and cardiac-muscle contraction. These processes are associated with muscle development. Furthermore, *QKI* was associated with dilated cardiomyopathy (DCM) and cardiac fibrosis [36]. Hence, it might play roles in cardiac health and disease [37]. We identified other essential pathways, such as focal adhesion and MAPK signaling. The former is related to cell proliferation, differentiation and cell activity, regulates gene expression and cell survival and participates in cell-membrane receptor and actin cytoskeleton construction [38]. Focal adhesion is an important signaling pathway in mammalian muscle development and skeletal-muscle regulation [39]. The DEG, *CAV-3*, is of great importance in skeletal-muscle development and regulation and controls the focal-adhesion signaling pathway. Furthermore, *CAV-3* is a muscle-specific protein that maintains muscle health and repairs damage [40]. The *CAV-3* knockout mice presented with a loss of membrane cellars in their muscle fibers and pathological changes in their skeletal muscle cells. Therefore, *CAV-3* is vital to muscle-tissue development.

The MAPK signaling pathway is ubiquitous and evolutionarily conserved and controls cell proliferation, motility, oxidative stress, survival and apoptosis [41,42]. The MAPK signaling cascade controls myocyte proliferation and differentiation by regulating the expression of myogenic TFs and the cell cycle. Furthermore, MAPK signaling also regulates skeletal-muscle movement and contraction [43]. The p38 signaling pathway is a member of the MAPK superfamily; it is expressed mainly in skeletal muscle and induces muscle-tissue differentiation. In addition, p38-activated phosphorylation is significantly upregulated when C2C12 myoblasts differentiate into myotubes [44]. Here, we demonstrated that *NFATC2* and *IGF1* were significantly enriched in the MAPK signaling pathway. The *NFATC2* isoforms are important for skeletal-muscle growth [45]. A member of the insulin-growth-factor family, *IGF1*, encodes a single-chain polypeptide hormone. This protein affects muscle development by regulating the synthesis of skeletal-muscle proteins. It plays vital roles in growth and metabolism [46]. Various *IGF1* isoforms generated by alternative splicing affect muscle-cell proliferation and differentiation [47]. Therefore, the associations of the aforementioned pathways and DEGs with the regulation of muscle growth and development merit further investigation.

## 5. Conclusions

In conclusion, we demonstrated that out of six different goat *QKI* isoforms, *QKI-5* had the highest expression level and the strongest ability to promote goat-myoblast differentiation. Hence, it was identified as the major isoform in this process. The overexpression of *QKI-5* and its subsequent induction of differentiation confirmed that *QKI* promotes goat-myoblast differentiation. The RNA-Seq identified 199 DEGs and the GO- and KEGG-enrichment analyses showed that *QKI* regulated multiple downstream genes and signaling pathways associated with muscle growth and development. However, the underlying molecular mechanisms by which *QKI* transcription regulates muscle development in goats are currently not clear and require further investigation.

## Figures and Tables

**Figure 1 animals-13-00725-f001:**
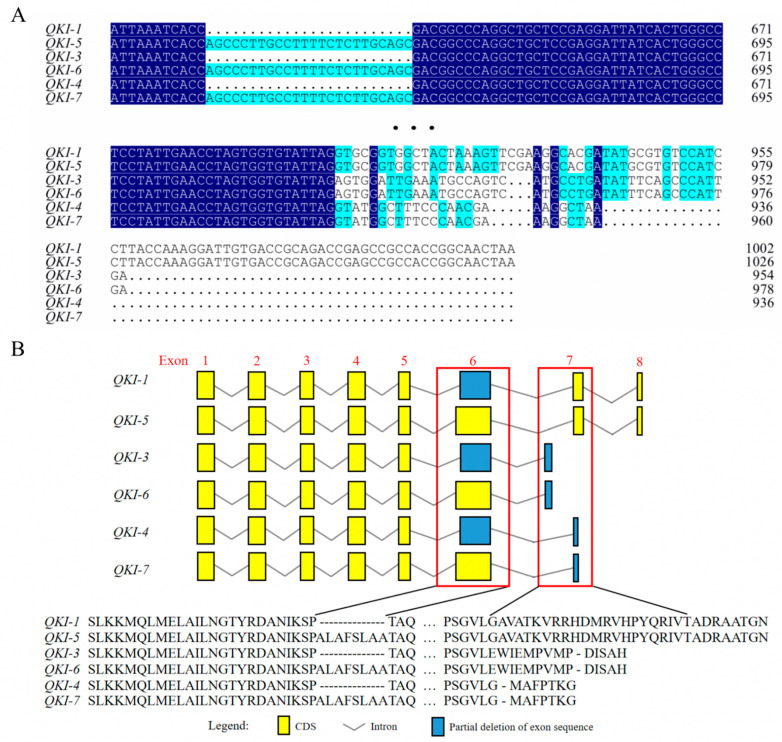
(**A**) Alignment of the amino-acid sequences on the different parts of the *QKI* isoforms. (**B**) Alternative splicing analysis of the *QKI* isoforms. Yellow rectangles indicate exons. Blue rectangles indicate partial deletion of exon sequences. Broken lines indicate introns. Red boxes indicate exon sites with possible alternative splicing.

**Figure 2 animals-13-00725-f002:**
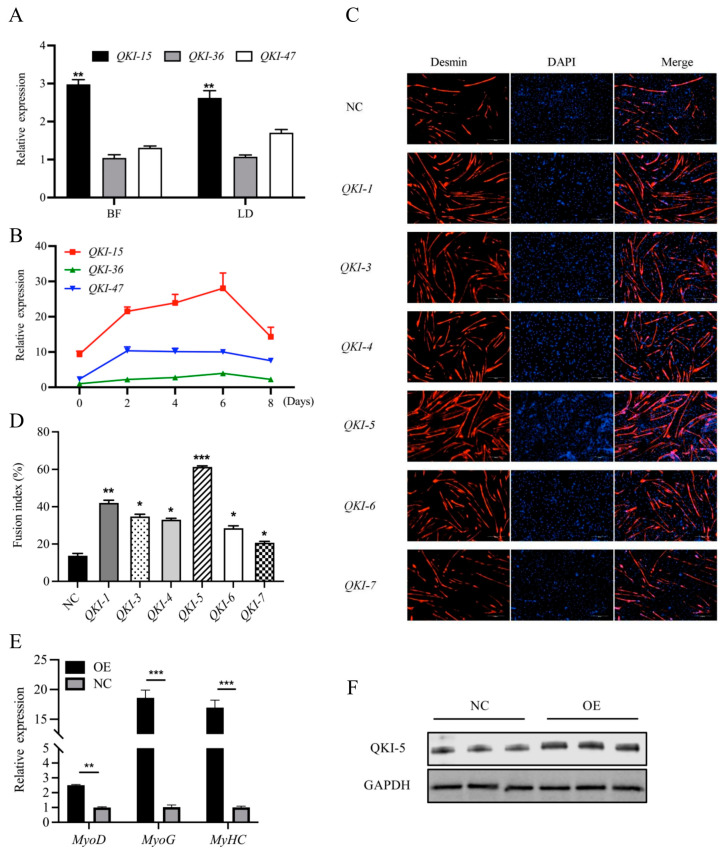
Expression levels of various *QKI* isoforms in goat-muscle tissue and myoblast differentiation. (**A**) The qPCR showing expression of different *QKI* isoforms in goat LD and BF muscle tissues. Data are means ± SEM; *n* = 6/group. (**B**) The qPCR showing expression trends of different *QKI* isoforms during goat-myoblast differentiation. (**C**) Immunofluorescence of goat-myotube differentiated for 6 d in presence of *QKI* overexpression or negative control (NC). Myotubes were fixed and stained with desmin (red) and DAPI (blue). Scale bar = 100 μm. (**D**) Fusion indices of myotubes from (**C**) calculated as % of nuclei within myotubes. Control group was NC (*** *p* < 0.001; ** *p* < 0.01; * *p* < 0.05). (**E**) Expression analysis of marker genes during myoblast differentiation. (**F**) Protein expression in *QKI* overexpression (OE) and negative control (NC) groups. GAPDH: internal reference gene.

**Figure 3 animals-13-00725-f003:**
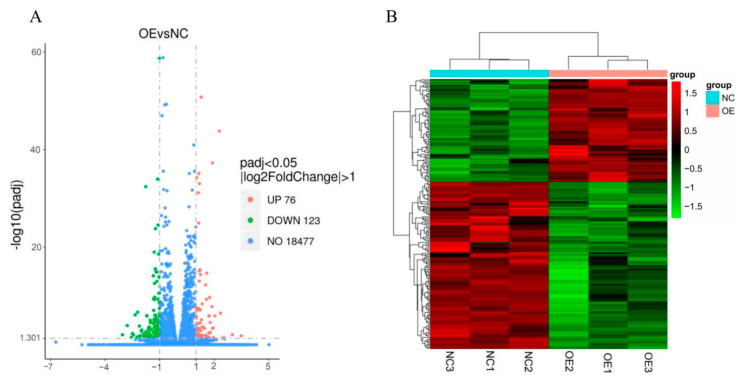
Analysis of DEGs between two groups of RNA-Seq data. (**A**) Volcano plots of DEGs between overexpressed *QKI* and negative control (NC). Red and green dots represent upregulated and downregulated genes, respectively, in OE group compared with NC group. (**B**) Heatmap of hierarchical cluster analysis of DEGs between samples. The horizontal coordinates are the sample names and the vertical coordinates are the normalized values of the differential gene FPKM. The expression level increases with the deepening of red and decreases with the deepening of green.

**Figure 4 animals-13-00725-f004:**
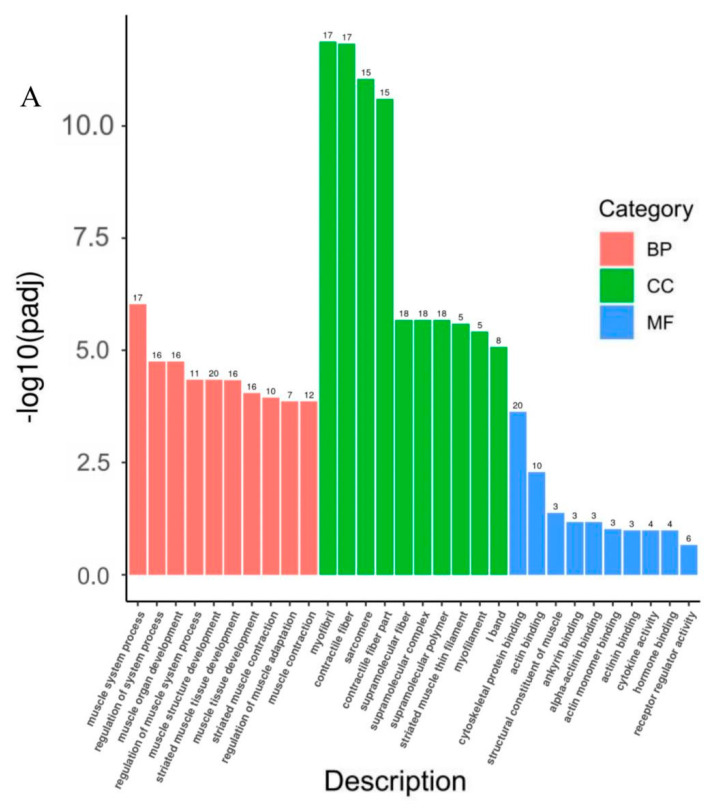
GO and KEGG pathway analyses of DEGs. (**A**) Gene ontology (GO) enrichment of DEGs. FDR ≤ 0.05 is considered significantly enriched. (**B**) Bubble chart of KEGG−function−enrichment analysis of DEGs. Abscissa is ratio of number of DEGs annotated to KEGG pathway to total number of DEGs. Ordinate is KEGG pathway. The KEGG node containing upregulated gene is marked in red. The KEGG node containing downregulated gene is marked in green. The KEGG node containing up− and downregulated gene is marked in yellow.

**Table 1 animals-13-00725-t001:** The qPCR primer sequences.

Primer Name	Forward	Reverse
*QKI-15*	ATGCCAAACGGAACTCCTCA	TAGCCACCGCACCTAATACAC
*QKI-36*	GCTGTCATGCCAAACGGAAC	ATCAGGCATGACTGGCATTTC
*QKI-47*	ATACCCCTACACGTTGGCAC	TCGTTGGGAAAGCCATACCT
*GAPDH*	GCAAGTTCCACGGCACAG	GGTTCACGCCCATCACAA
*MyoD*	GTGCAAACGCAAGACGACTA	GCTGGTTTGGGTTGCTAGAC
*MyoG*	GGACCCTACAGATGCCCACAA	TTGGTATGGTTTCATCTGGG
*MyHC*	CCACATCTTCTCCATCTCTG	GGTTCCTCCTTCTTCTTCTC

**Table 2 animals-13-00725-t002:** Genes related to muscle development and affected by *QKI*.

Gene	Description	log2 (FC)	*p*-Value	Enriched Terms
ACTA1	actin, alpha 1, skeletal muscle	−1.002515693	0.001516326	1, 2, 3, 4, 6, 7, 8, 9, 10
ATP1A2	ATPase Na+/K+ transporting subunit alpha 2	−1.121160481	0.000335005	5
CACNG1	calcium voltage-gated channel auxiliary subunit gamma 1	−1.094915942	0.007555945	1, 4, 6, 7, 8, 10
CAV3	caveolin 3	−1.120522417	0.016390643	1, 6, 8, 9, 10
CHRNA1	cholinergic receptor nicotinic alpha 1 subunit	−1.302552151	7.09 × 10^−6^	1, 4, 5, 7
DES	desmin	−1.30009742	6.31 × 10^−6^	2, 3
HRC	Histidine-rich calcium-binding protein	−1.12825787	0.000613572	2, 3
IGF1	insulin-like growth factor I	−1.079802091	2.91 × 10^−25^	1, 4, 6, 8
JPH1	junctophilin 1	−1.013695004	0.018021031	1, 2, 3, 4
LDB3	LIM domain binding 3	−1.09115513	4.54 × 10^−10^	1, 2, 3, 6, 8, 10
NFATC2	nuclear factor of activated T cells 2	3.004126141	0.008888678	1, 6, 8, 9
SCX	scleraxis bHLH transcription factor	−1.682899293	0.025556357	1, 4, 7
TNNC1	troponin C1, slow skeletal and cardiac type	−1.251575045	1.19 × 10^−5^	1, 2, 3, 4, 5
TNNC2	capra hircus troponin C2, fast skeletal type (TNNC2), mRNA.	−1.333657493	4.12 × 10^−10^	2, 3, 5
TNNI1	troponin I1, slow skeletal type	−1.280560498	2.66 × 10^−13^	1, 4
TNNT1	troponin T1, slow skeletal type	−1.762686865	3.90 × 10^−33^	2, 3, 5
TNNT3	troponin T3, fast skeletal type	−1.60504666	0.000335191	2, 3, 5
VGLL2	vestigial like family member 2	−1.006900776	0.003217405	1, 4, 7
XIRP2	xin actin binding repeat containing 2	−1.210101158	0.000207475	1, 2, 3, 4
CYP26B1	cytochrome P450 family 26 subfamily B member 1	1.352173933	0.047638797	1, 4, 6, 7, 9

1, muscle-structure development; 2, myofibril; 3, contractile fiber; 4, muscle-organ development; 5, muscle contraction; 6, muscle-cell differentiation; 7, skeletal-muscle-tissue development; 8, muscle-cell development; 9, myotube differentiation; 10, myofibril assembly.

## Data Availability

All original PacBio- and Illumina-sequencing data in this study are accessible with the following link: https://dataview.ncbi.nlm.nih.gov/object/PRJNA917124?reviewer=ao9d5clg6r1tgh3lnbl4ptn33c (accessed on 1 January 2023).

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
