# Peer review of "Overexpression of the QKI Gene Promotes Differentiation of Goat Myoblasts into Myotubes"

_animals, 2023, doi:10.3390/ani13040725_

Round 1
Reviewer 1 Report
The manuscript titled "Overexpression of the QKI Gene Promotes Differentiation of Goat Myoblasts into Myotubes" demonstrated that overexpression of QKI gene promotes the differentiation of goat myoblasts into myotubes by changing the signal pathway of muscle growth and development. The information in this article may be helpful to study the goat skeletal muscle development. I have a few comments that may help authors improve the manuscript.
1. There are two Data Availability Statements in the manuscript, line 456 and 460 respectively. Two statements should be combined.
2. In Figure 3B, the number of down regulated genes is 123, but the number of down regulated genes written in the manuscript is 133. Please correct this error.
3. Line 275, There are errors in (| log2 (FoldChange) | ≥ 1 writing.
4. Please add the content represented by the Y-axis in Figure 4A and enlarge the font in Figure 4.
5. Please check the description of Line 281, there is no Figure 3C in the figure.
6. Please add what the red and green values represent in Figure 3A.
7. The overexpressed QKI genes in this article are six isoforms of QKI, and only the protein expression of QKI-5 is detected in Figure 2F. Please provide QKI1, 3, 4, 6, 7 protein results.
Author Response
亲爱的审稿人
我谨代表所有投稿作者,对您的来信和审稿人对我们题为“Overexpression of the QKI Gene Promotes Differentiation of Goat Myoblasts into Myotubes”的文章提出的建设性意见表示由衷的感谢。这些意见都很有价值,有助于改进我们的文章。根据审稿人的意见,我们对稿件进行了大量修改。请参阅下面的红色部分,以了解对审稿人的意见和疑虑的逐点回应。我们还要感谢您允许我们重新提交手稿的修订副本。我们希望修改后的手稿被接受在动物杂志上发表。
此致,
刘文中

Reviewer 2 Report
The manuscript by Chen et al elaborate on expression patterns of various goat QKI isoforms and reported QKI-5 the highest expression level and the strongest ability to promote goat myoblast differentiation.Further with RNA-sequencing 199 differentially expressed genes were identified and with enrichment analysis authors identified several signaling pathways and genes associated with muscle growth and development, which in future could be pivotal in developing novel goat lines with enhanced musculature and, by extension superior meat yield and quality, breeder profitability, and consumer satisfaction. Overall, the article is well written and very clear in its content- Figures, Tables.
I have a few minor comments and suggestions for authors to consider -
Line 279- Correct ‘Figure 3B’ to ‘Figure 3A’
Line 281- Correct ‘Figure 3C’ to ‘Figure 3B’
Figure 3- The figure description requires rewriting/rearrange. Currently under this figure, I guess 3B is a Volcano plot and 3A is Heatmap. Please check through manuscript text too (Line 279, 281) and correct it respectively.
Line 294-298 and footer notes of Table 2 state the same information. I suggest authors can avoid this repetitiveness , just by rewriting Line 294-298 - ‘ The obtained 20 DEGs associated with various selected GO terms as depicted in Table 2’
Figure 4 - For easy to refer and readability I suggest vertical representation for figure 4A - Gene ontology (GO) enrichment of 314 DEGs.
As final comments, the manuscript fits the journal’s scope, it has good English quality and organization. I recommend this manuscript for publication with minor revisions expressed above.
Author Response
Dear Reviewer
On behalf of all the contributing authors, I would like to express our sincere appreciations of your letter and reviewers’ constructive comments concerning our article entitled “Overexpression of the QKI Gene Promotes Differentiation of Goat Myoblasts into Myotubes”. These comments are all valuable and helpful for improving our article. According to the reviewers’ comments, we have made extensive modifications to our manuscript. Please see below, in red, for a point-by-point response to the reviewers’ comments and concerns. We would like also to thank you for allowing us to resubmit a revised copy of the manuscript. We hope that the revised manuscript is accepted for publication in the Animals.
Sincerely,
Wenzhong Liu

Round 2
Reviewer 1 Report
No more comments